# Critical Thermodynamic Conditions for the Formation of *p*-Type β-Ga_2_O_3_ with Cu Doping

**DOI:** 10.3390/ma14185161

**Published:** 2021-09-08

**Authors:** Chuanyu Zhang, Zhibing Li, Weiliang Wang

**Affiliations:** State Key Laboratory of Optoelectronic Materials and Technologies, Guangdong Province Key Laboratory of Display Material and Technology, School of Physics, Sun Yat-sen University, Guangzhou 510275, China; zhangchy26@mail2.sysu.edu.cn (C.Z.); stslzb@mail.sysu.edu.cn (Z.L.)

**Keywords:** gallium oxide, *p*-type semiconductor, thermodynamic conditions, first principle

## Abstract

As a promising third-generation semiconductor, ${\beta \text{-}\mathrm{Ga}}_{2}O_{3}$ is facing bottleneck for its p-type doping. We investigated the electronic structures and the stability of various Cu doped structures of ${\beta \text{-}\mathrm{Ga}}_{2}O_{3}$. We found that Cu atoms substituting Ga atoms result in *p*-type conductivity. We derived the temperature and absolute oxygen partial pressure dependent formation energies of various doped structures based on first principles calculation with dipole correction. Then, the critical thermodynamic condition for forming the abovementioned substitutional structure was obtained.

## 1. Introduction

Wide-bandgap semiconductors, also known as third-generation semiconductors, are the future trend of material development. As a popular type of these semiconductors, Gallium Oxide (${\beta \text{-}\mathrm{Ga}}_{2}O_{3}$) features a direct band gap of 4.9 eV [1], an absorption edge of 250 nm [2], and a breakdown field of 8 MV/cm [3], which indicates a promising candidate in the fields of light-emitting diodes [4], deep-ultraviolet detectors [5], and the integrated circuit [6,7]. While theoretical and experimental studies on *n*-type ${\beta \text{-}\mathrm{Ga}}_{2}O_{3}$ are relatively sufficient [1,8,9], the *p*-type doping is far away from successful, leading to major impediment for application, which attracts great attention of researchers.

In order to obtain ideal *p*-type doping of ${\beta \text{-}\mathrm{Ga}}_{2}O_{3}$, researchers have tried a variety of impurity elements. Guo et al. studied the electronic structures of 10 non-metal elements doped ${\beta \text{-}\mathrm{Ga}}_{2}O_{3}$, such as C, N and F [10]; Kyrtsos et al. selected 9 metal element such as Li, Na, K as a single substitution for a Ga atom [11], and results of these 2 works showed that it was difficult to obtain the desired shallow acceptor dopant. Neal et al. carried out experiments on four elements doping: Si, Ge, Fe, Mg. It was found that the doping of Si and Ge could form shallow donors, while the doping of Fe and Mg formed deep acceptors [12]. Other similar studies also show that it is hardly possible to form *p*-type doping, when a simple doping configuration is tried [13].

In view of this, researchers began to investigate the possibility of co-doping. In ${\beta \text{-}\mathrm{Ga}}_{2}O_{3}$ crystal, there are two different Ga-substituted sites, in addition to three different interstitial sites, which indicates very diverse doped structures, especially when there are more than one impurity atoms. Ma et al. studied five structures of Al-N, In-N co-doping, and N doping, and found that the acceptor level of ${\mathrm{In}}_{\mathrm{Ga}}-2N_{O}$ structure was the lowest [14]. Li et al. investigated eight configurations of N-P co-doping and N or P doping, and found that their defect energy level positions were obviously different, among them the $N1P2_{{\mathrm{Ga}}_{2}O_{3}}$ configuration had the most potential to form *p*-type [15].

In the above N-P co-doping study, researchers also found that the gallium-rich environment was conducive to co-doping [15]. Stephan Lany studied the Si doping and Ge doping ${\beta \text{-}\mathrm{Ga}}_{2}O_{3}$, and found that doping concentrations and properties depended on temperature and oxygen partial pressure [16]. Researchers usually assess the thermodynamic stability by calculating formation energy [17] or binding energy [18], and it means that chemicals with lower formation energy (or higher binding energy) are easier to produce [19]. In the research of ZnO, facing the similar dilemma with ${\beta \text{-}\mathrm{Ga}}_{2}O_{3}$, Hou et al. calculated formations of 16 structures of ZnO with Li doping to find the most stable one [20]; Jin et al. studied eight structures of Li-doped ZnO, by considering the temperature and absolute oxygen partial pressure dependent chemical potential, and found the *p*-type structure and the required thermodynamic conditions, which is consistent with experiments [21].

Therefore, it is essential to consider different doping structures to find *p*-type structures and analyze their thermodynamic stability. It is necessary to find the condition (temperature and partial pressure of relevant gases) for these *p*-type structures to form, in order to be helpful for experimentalists. Cu is most likely suitable for doping ${\beta \text{-}\mathrm{Ga}}_{2}O_{3}$ because the size of the Cu ion is close to that of the Ga atom. In this paper, we use the density functional theory (DFT) to study Cu-doped ${\beta \text{-}\mathrm{Ga}}_{2}O_{3}$. We consider different substitutional sites, interstitial sites, and their coupling. We calculated their electronic structures to find the *p*-type structure. Then, we calculated their formation energies as functions of temperature and absolute oxygen partial pressure to compare the stabilities of these structures, and thus find the thermal dynamic condition for the *p*-type structure to form.

## 2. Theory and Computational Details

In this paper, the DFT calculations are performed with Vienna Ab initio Simulation Package (VASP) [22], using the Generalized Gradient Approximation (GGA) exchange−correlation functional [23], and Projector Augmented-Wave (PAW) method [24]. The cut-off energy for the plane-wave expansion is set to 520 eV. Full relaxation of magnetization was performed for spin-polarized calculations. All ions are relaxed till all forces acting on ions are smaller than 0.001 eV/Å and the total energy change between electronic self-consistent loops is smaller than $10^{-4}$ eV. We use Monkhorst–Pack mesh K-points that vary from 5 × 5 × 5 to 2 × 2 × 2 depending on supercell size, in structure optimization and static energy calculation. When calculating density of states, we use a 15 × 15 × 15 K-points for 1 × 2 × 2 supercell. The symmetry of doped structures is assumed to be P1.

### 2.1. Dopant Formation Energy

The formation energy of the doped structure is:(1)$$E_{f}{=E}_{d}-E_{p}{+n}_{\mathrm{Ga}}\left(E_{\mathrm{Ga}}{+\mu }_{\mathrm{Ga}}\right){-n}_{\mathrm{Cu}}\left(E_{\mathrm{Cu}}{+\mu }_{\mathrm{Cu}}\right),$$
where $n_{\mathrm{Ga}}$ ($n_{\mathrm{Cu}}$) is the number of Ga (Cu) atoms removed from (added to) the supercell. $E_{d}$ ($E_{p}$) is the total energy of the doped (pure) ${\beta \text{-}\mathrm{Ga}}_{2}O_{3}$. $E_{\mathrm{Ga}}$ ($E_{\mathrm{Cu}}$) is the average energy of each Ga (Cu) atom in bulk Ga (Cu). The above total energies and average energies are all calculated with DFT. Chemical potential of Ga atom $\mu_{\mathrm{Ga}}$ is determined by that of $O_{2}$ and ${\mathrm{Ga}}_{2}O_{3}$:(2)$$\mu_{\mathrm{Ga}}\left({T,p}_{O_{2}}\right)=\frac{1}{2}\mu_{{\mathrm{Ga}}_{2}O_{3}}\left({T,p}^{0}\right)-\frac{3}{4}\mu_{O_{2}}\left({T,p}_{O_{2}}\right),$$
where:(3)$$\mu_{O_{2}}\left({T,p}_{O_{2}}\right){=\mu }_{O_{2}}\left({T,p}^{0}\right){+k}_{B}T\ln\left(\frac{p_{O_{2}}}{p^{0}}\right),$$
where $p^{0}=1\;\mathrm{bar}$, $p_{O_{2}}$ is the absolute oxygen partial pressure.

$\mu_{{\mathrm{Ga}}_{2}O_{3}}\left({T,p}^{0}\right)$ and $\mu_{O_{2}}\left({T,p}^{0}\right)$ can be looked up in the references [25,26,27], coming from experiment measurement. $\mu_{\mathrm{Cu}}\left(T\right)$ can also be looked up in the references [25,26,27], as it varies with temperature. The chemical potential of Cu atom $\mu_{\mathrm{Cu}}$ is not relevant because we fixed $n_{\mathrm{Cu}\;}=2$ in all the doped supercells, which means there are always two Cu atoms in the supercell: either one substitutional Cu atom plus one interstitial Cu atom or two substitutional Cu atoms or two interstitial Cu atoms.

### 2.2. Electric Dipole Correction

As the periodic boundary condition is applied in the DFT calculation, the long-range electrostatic interactions among supercells will be counted into system energies, while this interaction in real doped structures should be close to zero because the dopant concentration is very low in the experiments. Therefore, it is necessary to correct the electric dipole interactions [21]. The interaction energy between two electric dipole moments is:(4)$$U=\frac{\vec{p}{}_{i}\cdot \vec{p}{}_{j}-3(\vec{p}{}_{i}\cdot \hat{r}{}_{ij})(\vec{p}{}_{j}\cdot \hat{r}{}_{ij})}{4\pi \varepsilon_{0}r_{ij}^{3}},$$
where $\vec{p}{}_{i}\cdot (\vec{p}{}_{j})$ is the electric dipole moment of the *i*-th (*j*-th) supercell, $\hat{r}{}_{ij}=\vec{r}/r$ is the unit vector pointing from dipole *i* to dipole *j*, $\varepsilon_{0}$ is the dielectric constant of vacuum. In our dipole correction work, we assume that the dipole in each supercell is the same ($\vec{p}{}_{i}=\vec{p}{}_{j}=\vec{p}$), and we calculate the sum of interaction energies among supercells in the 501 × 501 × 501 range around the central one, in order to correct the system energy.

## 3. Results and Discussion

Ga atoms in a ${\beta \text{-}\mathrm{Ga}}_{2}O_{3}$ cell are located at two different sites, which are named sites 1 and 2, respectively, and a Cu atom will be named ${\mathrm{Cu}}_{1}$ (${\mathrm{Cu}}_{2}$) when it substitutes at site 1 (2), as illustrated in Figure 1a,b. There are three different interstitial sites which are named sites A, B, and C, respectively, and an interstitial Cu atom is named ${\mathrm{Cu}}_{A}$, ${\mathrm{Cu}}_{B,}$ or ${\mathrm{Cu}}_{C},$ respectively, when the Cu atom occupies these interstitial sites. There are two Cu doping atoms near (N) or separated (S) with each other in each supercell.

Firstly, we calculated 36 doped structures in an 80-atom supercell, and selected 18 structures with lower energy to perform calculations in larger supercells. Then, we sorted them into three types: two substitutional atoms, one substitutional atom plus one interstitial atom, and two interstitial atoms. For further investigation, we chose the most stable structure in each type: ${\mathrm{Cu}}_{2}{+\mathrm{Cu}}_{2}-N$ (Figure 1c), ${\mathrm{Cu}}_{B}{+\mathrm{Cu}}_{1}-N$ (Figure 1e) and ${\mathrm{Cu}}_{C}{+\mathrm{Cu}}_{C}-N$ (Figure 1f).

### 3.1. Formation Energies vs. Supercell Size

We constructed 1 × 2 × 2, 1 × 3 × 2, 1 × 4 × 2, 2 × 3 × 2 and 2 × 4 × 2 supercells, which correspondingly consist of 80, 120, 160, 240, and 320 atoms. In the case of one substitutional Cu atom doping, the dopant concentrations are 1.25 at %, 0.83 at %, 0.62 at %, 0.42 at %, and 0.31 at %, respectively.

Figure 2 shows the formation energies of some structures at 0 K (i.e., $\mu_{\mathrm{Ga}}{=\mu }_{\mathrm{Cu}}=0$). Obviously, the formation energies perform an overall trend that decreases with the decrease of dopant concentrations in the low concentration region. The formation energy of interstitial doping changes more significantly with the concentration than the substitutional one, and their dipole correction energies are also larger, especially for the ${\mathrm{Cu}}_{C}{+\mathrm{Cu}}_{C}-N$ structure. After electric dipole correction, we can find that the formation energy tends to be constant in the low concentration region, which can be extended to the infinite size supercell.

### 3.2. Electronic Structure

We calculated the Band Structure (BS) and Projected Density of States (PDOS) of ${\mathrm{Cu}}_{2}{+\mathrm{Cu}}_{2}-N$, ${\mathrm{Cu}}_{B}{+\mathrm{Cu}}_{1}-N$ and ${\mathrm{Cu}}_{C}{+\mathrm{Cu}}_{C}-N$, as well as those of the intrinsic ${\beta \text{-}\mathrm{Ga}}_{2}O_{3}$. The calculated intrinsic band gap is 2.30 eV, which is consistent with previous theoretical results by standard DFT [28] and far from the experimental results [29], GGA + U [30,31] or hybrid DFT. It is very common that band gap is underestimated by DFT, and it can be improved by hybrid exchange-correlation functional or exchange-correlation energy whose derivative of the density (or density matrix) with respect to the number of electrons is discontinued at integers [32]. Our work focused on the relative positions of defect levels among different structures of Cu doped ${\beta \text{-}\mathrm{Ga}}_{2}O_{3}$, which always show a similar conductivity type when different methods are used [33,34].

As shown in Figure 3a, the VBM of intrinsic ${\beta \text{-}\mathrm{Ga}}_{2}O_{3}$ is mainly contributed by O-2p orbit, while the VBMs (defect levels) of the doped structure (${\mathrm{Cu}}_{2}{+\mathrm{Cu}}_{2}-N$) are mainly contributed by Cu-3d and O-2p orbit (Figure 3b). Pure ${\beta \text{-}\mathrm{Ga}}_{2}O_{3}$ is nonmagnetic. According to the PDOS (Figure 3b) and BS (Figure 4) of the ${\mathrm{Cu}}_{2}{+\mathrm{Cu}}_{2}-N$ structure, there are four spin down defect levels above the Fermi level near the VBM. These defect levels are acceptors, which implies the *p*-type conductivity.

Figure 3c shows PDOS of the ${\mathrm{Cu}}_{B}+{\mathrm{Cu}}_{1}-N$ structure. It has four defect levels of spin up (all occupied) and four of spin down (two occupied). We can find many defect levels that almost occupy the whole forbidden band, among which the one with the highest energy is only 0.223 eV away from the CBM. So, it is impossible to form *p*-type or *n*-type conductivity in this structure, due to the compensation between acceptors and donors. As for the ${\mathrm{Cu}}_{C}{+\mathrm{Cu}}_{C}-N$ structure (Figure 3d), which is nonmagnetic, the Fermi level is raised to the conduction band, which shows that this structure is *n*-type.

### 3.3. Critical Thermodynamic Conditions

Through the calculation of the electronic structures, we found that the *p*-type conductivity can be obtained only in the ${\mathrm{Cu}}_{2}{+\mathrm{Cu}}_{2}-N$ structure. Therefore, it is of great significance to calculate and analyze the thermodynamic stability of these structures in different conditions. We calculated the formation energies of ${\mathrm{Cu}}_{2\;}{+\mathrm{Cu}}_{2}-N$, ${\mathrm{Cu}}_{B}{+\mathrm{Cu}}_{1}-N$ and ${\mathrm{Cu}}_{C}{+\mathrm{Cu}}_{C}-N$ structures, as a function of the absolute oxygen partial pressure, at every 100 °C from 300 °C to 1500 °C, which is shown in Figure 5.

As shown in Figure 5, at very low absolute oxygen partial pressure, the formation energy of ${\mathrm{Cu}}_{C\;}{+\mathrm{Cu}}_{C}-N$ (red line) is the lowest. ${\mathrm{Cu}}_{B}{+\mathrm{Cu}}_{1}-N$ (black line) is the most stable in a large range near the standard atmospheric pressure, which indicates that it will become the dominant product under regular growth conditions. When the absolute oxygen partial pressure is high enough, the ${\mathrm{Cu}}_{2\;}{+\mathrm{Cu}}_{2}-N$ structure (blue line) is formed. It is owed to the changes of chemical potential of Ga and $O_{2}$. At a constant temperature, $\mu_{O_{2}}$ increases with the increase of absolute oxygen partial pressure (from the $k_{B}T\ln\left(p_{O_{2}}/p^{0}\right)$ term in Equation (3), and it means $\mu_{\mathrm{Ga}}$ decreases, which prompts the substitutional doping.

The intersection points of different lines in Figure 5 indicates the critical absolute oxygen partial pressure, at which the dominant product changes from a structure to another. Figure 6 shows critical oxygen partial pressures as a function of temperature. It is a phase diagram of ${\mathrm{Cu}}_{2\;}{+\mathrm{Cu}}_{2}-N$ (*p*-type), ${\mathrm{Cu}}_{B}{+\mathrm{Cu}}_{1}-N$ (*I*-type) and ${\mathrm{Cu}}_{C}{+\mathrm{Cu}}_{C}-N$ (*n*-type). *I*-type semiconductor means the semiconductor in which the acceptors and donors compensate each other. The critical oxygen partial pressure between ${\mathrm{Cu}}_{2}{+\mathrm{Cu}}_{2}-N$ and ${\mathrm{Cu}}_{B\;}{+\mathrm{Cu}}_{1}-N$ increases and then decrease with temperature due to two competitive effects: (1) The difference between the standard chemical potential of ${\mathrm{Ga}}_{2}O_{3}$ and $O_{2}$ ($\frac{1}{2}\mu_{{\mathrm{Ga}}_{2}O_{3}}\left({T,p}^{0}\right)-\frac{3}{4}\mu_{O_{2}}\left({T,p}^{0}\right)$) and thus the chemical potential of Ga ($\mu_{\mathrm{Ga}}$) increases with temperature, which suppresses (promotes) substitutional (interstitial) doping at higher temperatures; (2) On the other hand, the chemical potential of $O_{2}$ increases significantly with temperature when the absolute oxygen partial pressure is larger than 1 bar due to the $k_{B}T\ln\left(p_{O_{2}}/p^{0}\right)$ term in Equation (3), which promotes (suppresses) substitutional (interstitial) doping in higher temperature. The former effect is more significant than the latter when the temperature is lower than 900 K. Therefore, the critical oxygen partial pressure between ${\mathrm{Cu}}_{2\;}{+\mathrm{Cu}}_{2}-N$ and ${\mathrm{Cu}}_{B\;}{+\mathrm{Cu}}_{1}-N$ increases with temperature when the temperature is lower than 900 K. The latter effect is more significant than the former when the temperature is higher than 900 K. Therefore, the critical oxygen partial pressure between ${\mathrm{Cu}}_{2}{+\mathrm{Cu}}_{2}-N$ and ${\mathrm{Cu}}_{B}{+\mathrm{Cu}}_{1}-N$ decreases with temperature when the temperature is higher than 900 K. This nonmonotonic critical oxygen partial pressure is distinctly different from the monotonic critical oxygen partial pressure for *p*-type ZnO formed with Li doping [21]. The standard chemical potential of ${\mathrm{Ga}}_{2}O_{3}$ and $O_{2}$, as well as critical oxygen partial pressures, are listed in Table 1. The reaction condition of 300 °C and absolute oxygen partial pressure of 802 bar is the most convenient condition to obtain the *p*-type structure, without considering the reaction rate.

This absolute oxygen partial pressure is much higher than that of conventional ${\beta \text{-}\mathrm{Ga}}_{2}O_{3}$ growth [1], and much lower than that of phase equilibrium pressure between β-and ${\beta \text{-}\mathrm{Ga}}_{2}O_{3}$ [35].

## 4. Conclusions

Based on DFT, we calculated Cu-doped ${\beta \text{-}\mathrm{Ga}}_{2}O_{3}$ supercells, containing from 80 to 320 atoms, and used dipole correction to eliminate the influence of long-range electrostatic interaction. Considering different substitutional and interstitial sites and distances of impurity atoms, we found that ${\mathrm{Cu}}_{2}{+\mathrm{Cu}}_{2}-N$ structure is a *p*-type semiconductor through the analysis of band structure and PDOS.

The stabilities of different doped structures depend on temperature and absolute oxygen partial pressure. We calculated the formation energies in different conditions, and found that the ${\mathrm{Cu}}_{B}{+\mathrm{Cu}}_{1}-N$ structure, which is a *I*-type, is the easiest to form in non-extreme conditions. By calculating the critical absolute oxygen partial pressure at which the ${\mathrm{Cu}}_{B\;}{+\mathrm{Cu}}_{1}-N$ structure transforms to the ${\mathrm{Cu}}_{2}{+\mathrm{Cu}}_{2}-N$ one, we can conclude that an extremely high absolute oxygen partial pressure is required to make the *p*-type ${\mathrm{Cu}}_{2}{+\mathrm{Cu}}_{2}-N$ structure the dominant product. A relatively easier reaction condition is the absolute oxygen partial pressure of 802 bar at the temperature of 300 K without considering the reaction rate.

## Figures and Tables

**Figure 1 materials-14-05161-f001:**
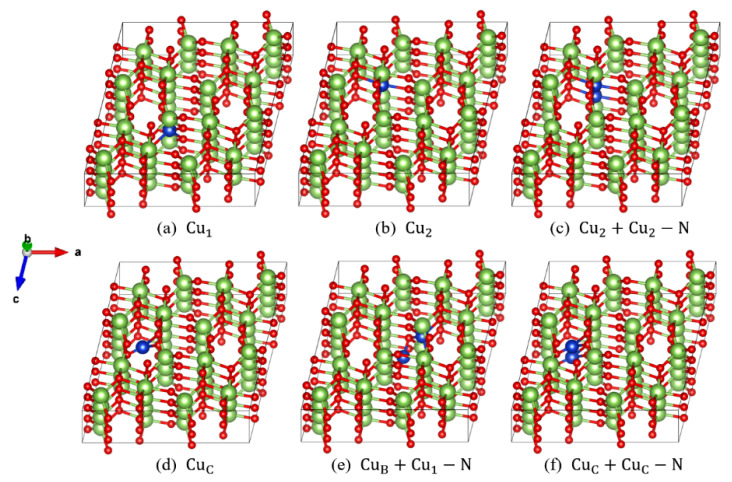
The most stable and representative doping structures: (**a**) ${\mathrm{Cu}}_{1}$, (**b**) ${\mathrm{Cu}}_{2}$, (**c**) ${\mathrm{Cu}}_{2}{+\mathrm{Cu}}_{2}-N$, (**d**) ${\mathrm{Cu}}_{C}$, (**e**) ${\mathrm{Cu}}_{B\;}{+\mathrm{Cu}}_{1}-N$ and (**f**) ${\mathrm{Cu}}_{C}{+\mathrm{Cu}}_{C}-N$. The red, green, and blue balls represent O, Ga, and Cu atoms, respectively.

**Figure 2 materials-14-05161-f002:**
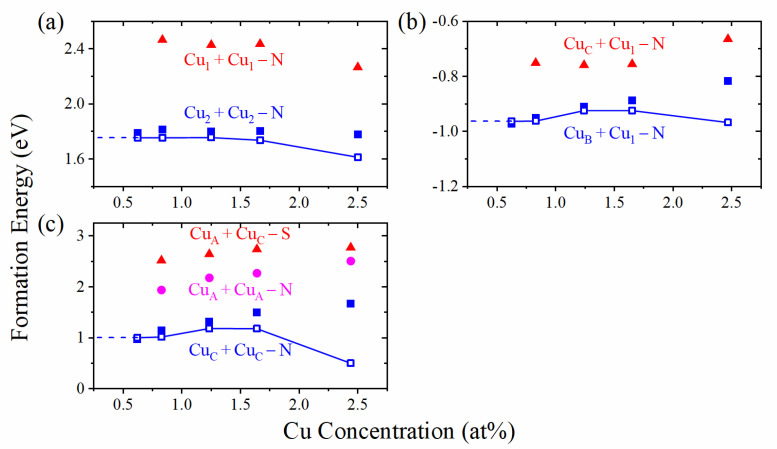
Formation energies at 0 K with (open) or without (filled) dipole correction of two substitutional Cu doped structures (**a**), one substitutional plus one interstitial Cu doped structures (**b**), and two interstitial Cu doped structures (**c**). Those of the structures with higher energy are shown without dipole correction. The dotted line represents extending to the concentration of 0%.

**Figure 3 materials-14-05161-f003:**
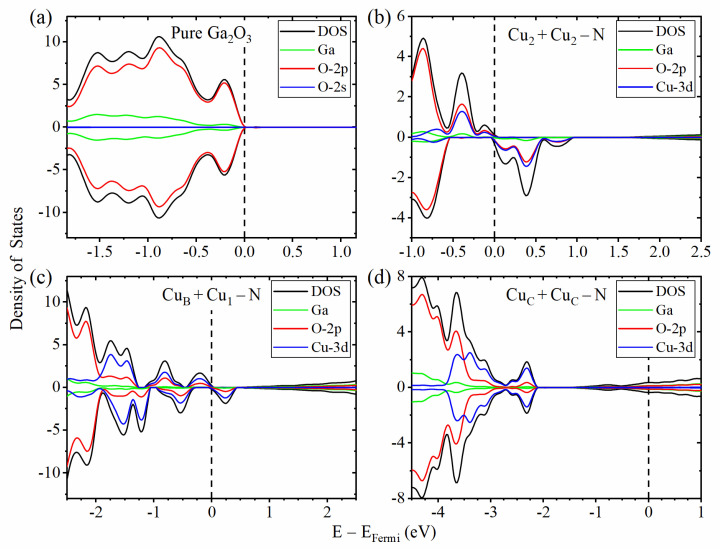
The PDOS of pure ${\beta \text{-}\mathrm{Ga}}_{2}O_{3}$ (**a**) and doped ${\beta \text{-}\mathrm{Ga}}_{2}O_{3}$: ${\mathrm{Cu}}_{2}\;+\;{\mathrm{Cu}}_{2}-N$ (**b**), ${\mathrm{Cu}}_{B}\;+\;{\mathrm{Cu}}_{1}-N$ (**c**) and ${\mathrm{Cu}}_{C}\;+\;{\mathrm{Cu}}_{C}-N$ (**d**). The dashed line represents the Fermi level, the black solid line represents the total density of states, the green and red lines represent the projected density of states of Ga and O-2p orbits, respectively. The blue lines represent O-2s orbit in Figure (**a**) and represent Cu-3d orbits in Figure (**b**–**d**). The positive (negative) DOS is the DOS with spin up (down).

**Figure 4 materials-14-05161-f004:**
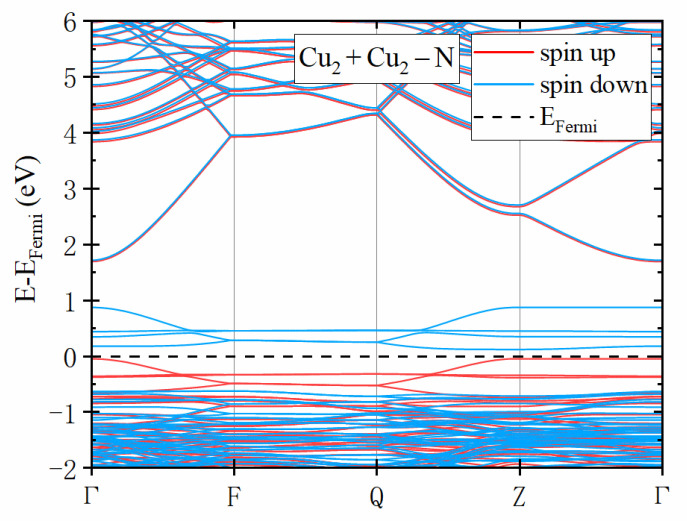
The band structure of the ${\mathrm{Cu}}_{2}{+\mathrm{Cu}}_{2}-N$ structure. The red and blue lines represent two spin states, and the dashed line represents the Fermi level.

**Figure 5 materials-14-05161-f005:**
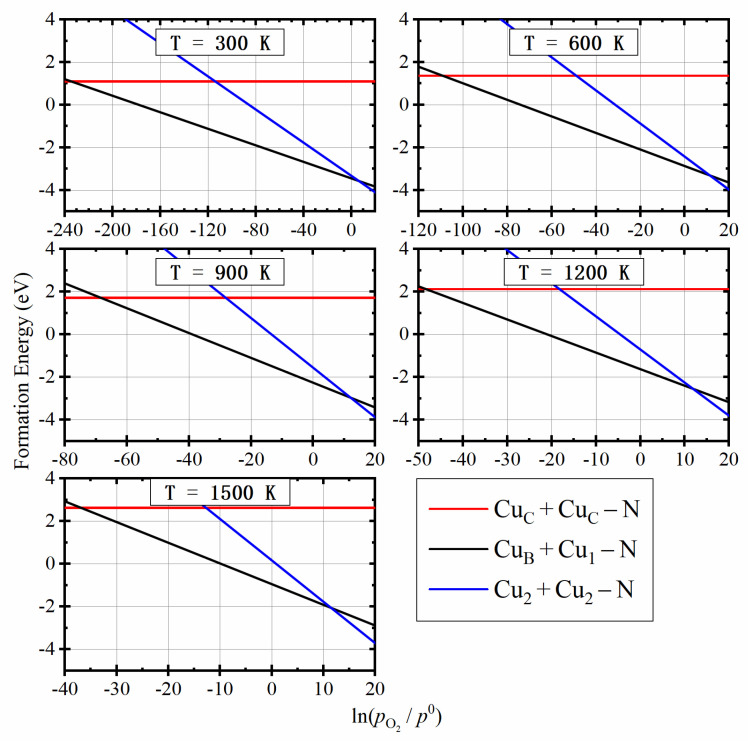
Formation energies of the three most stable structures as the function of absolute oxygen partial pressure at different temperatures.

**Figure 6 materials-14-05161-f006:**
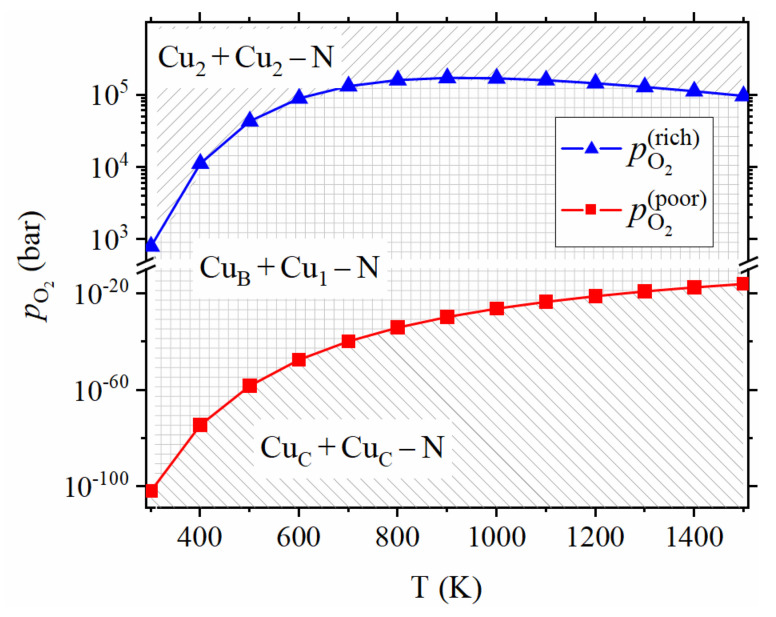
The phase diagram of Cu doped ${\beta \text{-}\mathrm{Ga}}_{2}O_{3}$. Blue dots and lines represent the critical absolute oxygen partial pressure between ${\mathrm{Cu}}_{B}{+\mathrm{Cu}}_{1}-N$ and ${\mathrm{Cu}}_{2}{+\mathrm{Cu}}_{2}-N$ structures, corresponding to the intersection point of black and blue lines in Figure 5, and it is named $p_{O_{2}\;}^{\left(\mathrm{rich}\right)}$. Red dots and lines represent the critical absolute oxygen partial pressure between ${\mathrm{Cu}}_{C\;}{+\mathrm{Cu}}_{C}-N$ and ${\mathrm{Cu}}_{B\;}{+\mathrm{Cu}}_{1}-N$, corresponding to the intersection point of red and black lines in Figure 5, and it is named $p_{O_{2}\;}^{\left(\mathrm{poor}\right)}$.

**Table 1 materials-14-05161-t001:** The standard chemical potential of ${\mathrm{Ga}}_{2}O_{3}$ and $O_{2}$, and critical absolute oxygen partial pressures of structure transition between ${\mathrm{Cu}}_{B}{+\mathrm{Cu}}_{1}-N$ and ${\mathrm{Cu}}_{2}{+\mathrm{Cu}}_{2}-N$ ($p_{O_{2}\;}^{\left(\mathrm{rich}\right)}$ ), between ${\mathrm{Cu}}_{C\;}{+\mathrm{Cu}}_{C}-N$ and ${\mathrm{Cu}}_{B}{+\mathrm{Cu}}_{1}-N$ ($p_{O_{2}\;}^{\left(\mathrm{poor}\right)}$ ).

T/K.	$\mu_{{\mathrm{Ga}}_{2}O_{3}}/\mathrm{eV}$	$\mu_{O_{2}}/\mathrm{eV}$	$p_{O_{2}}^{\left(\mathrm{poor}\right)}/\mathrm{bar}$	$p_{O_{2}}^{\left(\mathrm{rich}\right)}/\mathrm{bar}$
300	−11.556	−0.638	1.45 × 10^−102^	802
400	−11.660	−0.856	3.10 × 10^−75^	1.12 × 10^4^
500	−11.790	−1.081	6.14 × 10^−59^	4.30 × 10^4^
600	−11.944	−1.313	3.79 × 10^−48^	8.91 × 10^4^
900	−12.515	−2.039	2.08 × 10^−30^	1.71 × 10^5^
1200	−13.218	−2.802	9.43 × 10^−22^	1.45 × 10^5^
1500	−14.022	−3.593	1.09 × 10^−16^	9.65 × 10^4^

## Data Availability

The data presented in this study are available on request from the corresponding author.

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
