# Peer review of "Critical Thermodynamic Conditions for the Formation of p-Type β-Ga2O3 with Cu Doping"

_materials, 2021, doi:10.3390/ma14185161_

Round 1
Reviewer 1 Report
Although the results seem novel and the manuscript is generally well-written there are some important issues that should be clarified, as follows:
1. The well-known fact that oxide materials should be studied with methods beyond the standard GGA approach is mentioned in the manuscript. The results presented in this work may be strongly biased by the lack of adequate exchange correlation functional. The GGA+U, hybrid DFT (HSE etc.) and metaGGA (MBJLDA) methods are generally accepted in the literature, e.g.:
[A] C. Zhang et al., Electronic transport properties in metal doped β-Ga2O3: A first principles study, Phys. B 562 (2019) 124;
[B] S. Gao et al., Effect of transition metals doping on electronic structure and optical properties of β-Ga2O3, Mater. Res. Express 8 (2021) 025904.
What is the correspondence between the results presented in this work and the literature data obtained with the better approaches?
2. The list of references is incomplete. Other DFT-based reports on Cu-doped Ga2O3 should be cited in the present work.
3. The zero energy in Fig. 3 (a) should be set at Fermi energy (VBM).
Author Response
Thank you very much for your attention and comments on our paper. I have revised the manuscript according to your kind advices and detailed suggestions.
Please see the attachment.

Reviewer 2 Report
The paper is not very well written and organized to meet the publication standards. I don't think that the current version of the manuscript can be accepted. The topic of research is interesting, but the selected computational approach and results raise questions. The following issues need to be addressed:
- How well structure of the crystal is reproduced by selected computational scheme? What is the symmetry of your defect systems, is it just P1? Especially, when substituting Ga atoms, is symmetry reduced or all symmetry operators of perfect crystal are preserved?
- Electric dipole correction. Reference 28 is all about correction schemes for charged defect calculation, it does not mention electric dipole correction and does not contain Eq. 4. Please provide correct reference. From fig. 2 we can see that applied correction doesn't help to achieve constant formation energy for all concentrations, even the opposite - correction for small supercells underestimates the formation energy. Looking at largest supercells it can be suggested, that this correction is unnecessary - it almost does not affect formation energies in 160-, 240-, 320-atoms supercells
- Fig.3 and PDOS. How do you define the Fermi level for each case? You mention that "Cu2+Cu2-N structure, there are four spin down defect levels above VBM", but it seems that valence band maximum of Ga2O3 is around -0.6eV on Fig.3b, then there are defect levels of 4 unpaired electrons (2 Cu2+ and 2 O-, probably) and only then hole defect levels. Does this change the conclusions on defect type? It is worth mentioning the spin states of other systems for more clarity. Also, how will the total energy change if you consider total spin of supercell = 0 (each Cu2+ and O- have opposite spins).
- Probably, it would be easier for reader to understand Fig.3 if the 0 eV point was fixed at VBM of pristine crystal.
- Band gap problem. If the defect levels are in the band gap at 2.5 eV and higher above the VBM, while your calculation scheme gives band gap of pristine crystal only 2.3 eV, it is impossible to say either defect levels in real crystal mix with conduction band or do they just stay in the band gap, this would influence the further analysis of the defect. Can you improve this (maybe single-point calculation on higher level of theory (hybrid/+U)) or at least provide justification for your approach?
- For interstitial Cu atoms, if it is a neutral atom with electronic structure [Ar] 4s2, 3d9, are 4s-states shown on PDOS? Caption only mentions 3d orbitals.
- Is it reasonable to calculate properties at high temperature without taking into account phonon part, Gibbs free energy and other thermodynamic parameters? It was also shown (https://pubs.acs.org/doi/10.1021/acs.jpcc.0c08183) that at high pressures alpha phase is more stable, so the results of defect formation energies at high oxygen pressures become very questionable. Probably it also would be useful to add some sentences on crystal growth methods in Introduction, where your results could be most applicable.
- "at very low absolute oxygen partial pressure, the formation energy of CuC+CuC-N (red line) is the lowest." From Fig.5 this is only true for T >900 K. Why CuC+CuC-N formation energy is dependent on the temperature? Is chemical potential of copper in Eq.1 a function on the temperature or is it constant?
- "It is a phase diagram of Cu2+Cu2-N (p-type), CuB+Cu1-N (I-type) and CuC+CuC-N (n-type)." What does I-type mean?
- In your research you assume that formation of two copper substitutional defect would be best for p-type conductivity. Why it should be better than single CuGa? It would be also useful if you provide formation energies for single CuGa defects, and Cu2+Cu2 with different distances between two atoms, proving that clustering of these defects is thermodinamically favourable. Maybe it is worth adding your results ("36 doped structures in 80-atom supercell, and select 18 structures with lower energy to perform calculations in larger supercells.") in Supplementary.
Author Response

(The authors gave the same response as above.)

Round 2
Reviewer 2 Report
Thank You for detailed answers.
few small comments:
In first comment I was thinking that you mention cell parameters and maybe bond lengths of Ga2O3 just to show that they are in reasonable agreement with exp.data.
"We did not take phonon part into account because the number of atoms are the same for the dopped supercells we compared."
While it is probably true that phonon inclusion would have same effect on all three systems, but most likely it would also have effect on key values that you calculated (temperature and O2 pressure).
Author Response
Thank you very much for your comments again. I have revised the manuscript according to your kind advices and detailed suggestions.
Please see the attachment.
